

# PM$_{2.5}$ exposure aggravates kidney damage by facilitating the lipid metabolism disorder in diabetic mice

Yuecheng Jiang[1,2,3,*], Yanzhe Peng[3,*], Xia Yang[2,3,4], Jiali Yu[2,3,4], Fuxun Yu[2,3], Jing Yuan[3] and Yan Zha[1,2,3,4]

[1] Zunyi Medical University, Guiyang, China
[2] NHC Key Laboratory of Pulmonary Immunological Disease, Guizhou Provincial People's Hospital, Guiyang, China
[3] Department of Nephrology, Guizhou Provincial People's Hospital, Guiyang, China
[4] School of Medicine, Guizhou University, Guiyang, China
[*] These authors contributed equally to this work.

Corresponding author
Yan Zha, zhayan72@yeah.net,
zhayangz72@126.com

## ABSTRACT

**Background.** Ambient fine particulate matter $\leq 2.5\,\mu m$ (PM$_{2.5}$) air pollution exposure has been identified as a global health threat, the epidemiological evidence suggests that PM$_{2.5}$ increased the risk of chronic kidney disease (CKD) among the diabetes mellitus (DM) patients. Despite the growing body of research on PM$_{2.5}$ exposure, there has been limited investigation into its impact on the kidneys and the underlying mechanisms. Past studies have demonstrated that PM$_{2.5}$ exposure can lead to lipid metabolism disorder, which has been linked to the development and progression of diabetic kidney disease (DKD).

**Methods.** In this study, db/db mice were exposed to different dosage PM$_{2.5}$ for 8 weeks. The effect of PM$_{2.5}$ exposure was analysis by assessment of renal function, pathological staining, immunohistochemical (IHC), quantitative real-time PCR (qPCR) and liquid chromatography with tandem mass spectrometry (LC–MS/MS) based metabolomic analyses.

**Results.** The increasing of Oil Red staining area and adipose differentiation related protein (ADRP) expression detected by IHC staining indicated more ectopic lipid accumulation in kidney after PM$_{2.5}$ exposure, and the increasing of SREBP-1 and the declining of ATGL detected by IHC staining and qPCR indicated the disorder of lipid synthesisandlipolysis in DKD mice kidney after PM$_{2.5}$ exposure. The expressions of high mobility group nucleosome binding protein 1 (HMGN1) and kidney injury molecule 1 (KIM-1) that are associated with kidney damage increased in kidney after PM$_{2.5}$ exposure. Correlation analysis indicated that there was a relationship between HMGN1-KIM-1 and lipid metabolic markers. In addition, kidneys of mice were analyzed using LC–MS/MS based metabolomic analyses. PM$_{2.5}$ exposure altered metabolic profiles in the mice kidney, including 50 metabolites. In conclusion the results of this study show that PM$_{2.5}$ exposure lead to abnormal renal function and further promotes renal injury by disturbance of renal lipid metabolism and alter metabolic profiles.

## INTRODUCTION

Ambient fine particulate matter, comprising particles with aerodynamic diameters smaller than 2.5 μm ($PM_{2.5}$), is a leading health risk factor worldwide. According to a study by the Global Burden of Disease, $PM_{2.5}$ contributed to 4.14 million deaths globally in 2019 (*GBD 2019 Risk Factors Collaborators, 2020*). While some countries, such as China, have seen a significant decrease in $PM_{2.5}$ concentration in recent years, others have experienced an increase in $PM_{2.5}$ levels, as noted in the "State of Global Air Report 2020." In 2019, more than 90% of the global population was exposed to hazardous levels of environmental PM2.5 pollution (*Health Effects Institute, 2020*). In 2021, *Li et al. (2021)* study found that $PM_{2.5}$ can increase the risk of chronic kidney disease (CKD) among patients with DM. This suggests that $PM_{2.5}$ may be related to kidney damage in DM patients; however, the mechanisms by which $PM_{2.5}$ exposure causes kidney damage remain largely unaddressed.

Diabetic kidney disease (DKD), a major complication of DM, is the leading cause of CKD and end-stage renal disease (*Zhang et al., 2016*; *American Diabetes Association, 2020*). Abnormal lipid metabolism is an established and significant risk factor for DKD and implicated in DKD occurrence and development of the disease. The renal tubular epithelial cells are an important site affected by abnormal lipid metabolism (*Yang et al., 2018b*), abnormal lipid metabolism causes apoptosis and atrophy of renal tubular epithelial cells. Increased lipid synthesis and decreased lipid breakdown in renal tubular epithelial cells are important mechanisms underlying disrupted lipid metabolism (*Zhao et al., 2022*; *Fang et al., 2020*). Recent studies have also shown that $PM_{2.5}$ is a risk factor for abnormal lipid metabolism, $PM_{2.5}$ has further been found to cause dyslipidemia, non-alcoholic fatty liver disease (NAFLD), and increased fatty acid synthesis while decreasing lipolysis in mice in mice (*Xu et al., 2019*). It is suggested that $PM_{2.5}$ exposure can cause lipid metabolism disorder. In light of the various effects of $PM_{2.5}$ on individuals with diabetes, this study aimed to examine the metabolic disturbances induced by $PM_{2.5}$ in diabetic mice and compare their responses to exposure. Based on our hypothesis, exposure to $PM_{2.5}$ would lead to metabolic disturbances and worsen the adverse effects in diabetic patients.

Recent research has found that the high mobility group nucleosome binding protein 1 (HMGN1) is involved in chronic low-grade inflammation (*Yang et al., 2018a*). HMGN1 elevates kidney injury molecule 1 (KIM-1) expression and chronic low-grade inflammation in DKD J Yu, J Da, F Yu, J Yuan, Y Zha, 2022, unpublished data. Researchers have further found that KIM-1, a marker of tubular injury (*Ix & Shlipak, 2021*), can aggravate renal tubular lipid metabolism disorders in DKD (*Mori et al., 2021*; *Huang & Parikh, 2021*). The present study aims to investigate the role of $PM_{2.5}$ in kidney damage, especially lipid metabolism disorders and to determine whether the underlying mechanisms of action are associated with HMGN1-KIM-1 in the kidneys of DKD mice.

## MATERIALS & METHODS

### Animals

Seven-week-old male C57BL/6 mice and db/db mice were purchased from Changzhou Cavens Laboratory Animals Ltd. All mice were housed five per cage on a 12-hour light/dark cycle and in a climate-controlled environment and had free access to food and water. Animal testing is in accordance with the national guidelines for experimental animal health, and the animal protection and utilization committee of the People's Hospital of Guizhou Province approved the study [2022 (295)].

### $PM_{2.5}$ preparation

The $PM_{2.5}$ used in the study was purchased from the Standard Reference Materials and was certified by the National Institute of Standards and Technology (SRM2975). $PM_{2.5}$ was diluted in phosphate buffered saline (PBS) prior to use.

### $PM_{2.5}$ exposure

The mice were allowed to acclimatize for 1 week prior to the experiments. C57BL/6 mice ($n = 5$) comprised the control group (CON) and db/db mice were randomly assigned to 4 groups: no $PM_{2.5}$ exposure group (DKD, $n = 5$), low concentration $PM_{2.5}$ exposure group (DKD+L, $n = 5$), medium concentration $PM_{2.5}$ exposure group (DKD+M, $n = 5$), and high concentration $PM_{2.5}$ exposure group (H, $n = 5$) (*Faul et al., 2007*). The ear tags of each group of mice were recorded. $PM_{2.5}$ exposure dose of mice was 1.8 mg/kg, 5.4 mg/kg, and 16.2 mg/kg, respectively, based on a previous report (*Zhang et al., 2017*). The mice were exposed to PBS or $PM_{2.5}$ by intratracheal instillation thrice a week for 8 weeks.

### Sample collection

When the mice reached 16 weeks of age, they were housed individually in metabolic cages to collect 24 h urine samples. They were then euthanized via intraperitoneal injection of sodium pentobarbital. The eyeballs of the mice were extirpated and blood samples were collected. Serum and urine supernatant were collected by centrifugation and stored at −80 °C until analysis. Left kidneys were collected and preserved at −80 °C for a liquid chromatography with tandem mass spectrometry (LC-MS/MS) analysis. The right kidneys were used for paraffin section, frozen section and qPCR detection.

### Biochemical indicators analysis

The serum blood urea nitrogen (BUN) levels were measured using a Urea Assay Kit (Enzy Chrom, DIUR-500); serum creatinine (SRC) levels were measured using a Creatinine Kit (Nanjing Jiancheng Bioengineering Institute, C011-2-1), urea protein levels were measured using Urea Protein Kit (Nanjing Jiancheng Bioengineering Institute, C035-2-1), serum triglyceride levels were measured using a Triglyceride Assay Kit (ETGA-200; Enzy Chrom). and serum total cholesterol levels were measured with using a Cholesterol Assay Kit (E2CH-100; Enzy Chrom).

### Histological analysis

The kidney tissues were postfixed and embedded in paraffin or frozen in an OCT compound. The paraffinized sections were stained with hematoxylin and eosin (HE),

Periodic Acid–Schiff (PAS) or immunohistochemical (IHC) stains. HE staining used HE Stain Kit (G1120; Beijing Solarbio, Beijing, China), PAS staining used PAS Stain Kit. For the IHC, the tissues were sectioned and affixed onto coated slides before being subjected to deparaffinization and rehydration. Endogenous peroxidase was blocked by immersing the tissues in 3% $H_2O_2$ for 10 min. Two additional washes in ddH2O were performed to remove any remaining after which the sections were then rinsed under tap water. Subsequently, the sections were heated in a pressure cooker, boiled for 2 min and then cooled. The sections were then dried and a hydrophobic pen was used to draw marks around tissues before they were washed with PBS. The sections were incubated in blocking solution (5% bovine serum albumin in PBS) for 30 min. Next, primary antibodies ADRP (15294-1-AP, 1:200; Proteintech, Rosemont, IL, USA), SREBP-1 (14088-1-AP, 1:100; Proteintech, Rosemont, IL, USA), ATGL (55190-1-AP, 1:400; Proteintech, Rosemont, IL, USA), HMGN1 (11695-1-AP,1:200; Proteintech, Rosemont, IL, USA), KIM-1 (ab47635, 1:200; Abcam, Cambridge, UK) diluted in antibody diluent (1% bovine serum albumin in PBS) was added and incubated overnight at 4 °C in a wet chamber. On the next day, the antibody solution was removed and the sections were washed with wash buffer for three times. The sections were incubated with goat anti-mouse (ab6789, 1:2000; Abcam, Cambridge, UK) or goat anti-rabbit (ab7090, 1:300; Abcam, Cambridge, UK) for 45 min at 37 °C, before being rewashed for three times and incubated with DAB, cores were counterstained with hematoxylin, dehydrated and cover slipped. The average optical density value of DAB positive reactant was detected by Image-Pro Plus 6.0 software to reflect the expression of detected protein molecules in renal tissue. Oil Red O staining used the Modified Oil Red O Staining Kit (C0158S; Beyotime, Jiangsu, China). Photographs were captured using a digital slide scanner (GScan-20; Gcell Technologies, Gurgaon, India). The density of DAB was quantified using Image-Pro Plus 6.0 software. The expression levels of the factors in the kidney were quantified using average integrated optical density values in Image-Pro Plus 6.0 software.

## Preparing samples for the LC-MS/MS analysis

A total of 100 mg of each kidney sample was accurately weighed. The amount of the sample was placed into a 2 ml centrifuge tube, and 1,000 μL of tissue extract (75% 9:1 methanol: chloroform, 25% $H_2O$) was added to the 2 ml centrifuge tube. Three steel balls were added, ground at 50 Hz for 60 s in the tissue grinder, and the above procedure was repeated twice; room temperature ultrasound for 30 min, then ice bath for 30 min. After centrifuging for 10 min at 12,000 rpm and 4 °C, the supernatant was transferred to a new 2 ml centrifuge tube, concentrate, and dry. A total of 200 μL 50% acetonitrile solution should be accurately added with 2-Amino-3-(2-chloro- phenyl)-propionic acid (4 ppm) (stored at 4 °C) to redissolve the sample; the supernatant is filtered through a 0.22 μm membrane and transferred to the LC-MS detection bottle (*Warren et al., 2017*).

## Metabolomics analysis based on LC-MS/MS

The data were first converted to mzXML format using MSConvert in the ProteoWizard software package (v3.0.8789) (*French et al., 2015*) and processed using XCMS (*Navarro-Reig et al., 2015*) for feature detection, retention time correction and alignment. Metabolite

identification was based on mass accuracy (30 ppm) and MS/MS data that matched with databases in HMDB (*Wishart et al., 2018*) and KEGG. Data normalization was performed using the robust LOESS signal correction (QC-RLSC) (*Gagnebin et al., 2017*) to correct for systematic bias. After normalization, a maximum of 30% relative standard deviation (RSD) was retained in QC to ensure proper metabolite identification.

For statistical analysis of multivariate data, R language ropes package methods used were PCA, PLS-DA, and OPLS-DA. By using MetaboAnalyst, different metabolites were analyzed according to their pathways (*Xia & Wishart, 2011*), which combines results from powerful pathway enrichment analysis with the pathway topology analysis. For biological interpretation of higher-level systemic functions, the identified metabolites in metabolomics were mapped to KEGG pathways.

## Statistical analysis

The data were analyzed using SPSS 26 and were presented as the mean $\pm$ standard error of the mean. The Student's $t$-test was used to compare data between the two groups. Comparison among multiple groups and pairwise comparison was performed using one-way ANOVA and LSD test. Correlation analyses were performed using Pearson's correlation analysis. $P < 0.05$ indicated statistically-significant results. (* $p < 0.05$, *vs.* CON group; [#] $p < 0.05$, *vs.* DKD group.)

## RESULTS

### PM$_{2.5}$ exposure aggravates the decline of renal function and dyslipidemia in diabetic mice

The BUN, SRC and urinary albumin-to-creatinine ratio (UACR) were higher in the DKD group than in the CON group. The BUN, SRC and UACR in the DKD+L, DKD+M and DKD+H groups were higher than those in the DKD group, indicating that PM$_{2.5}$ exposure made mice more susceptible to severe renal disease (Figs. 1A–1C). The serum triglyceride and serum total cholesterol levels were higher in the DKD group than in the CON group. After PM2.5 exposure the serum triglyceride and serum total cholesterol levels in the DKD+L, DKD+M and DKD+H groups were higher than those in the DKD group, indicating that the PM$_{2.5}$ exposure also made the mice more susceptible to hyperlipidemia (Figs. 1D–1E).

### PM$_{2.5}$ exposure worsens renal lipid metabolism disorder and kidney damage in diabetic mice

The histological analysis of the HE and PAS-stained kidney sections revealed a mild glomerular lesion consisting of mesangial matrix expansions and more extensive vacuolization in proximal tubular cells compared with those in the DKD group, but this was not observed in the CON group. Tubular damage was found to have increased in the DKD+L, DKD+M and DKD+H groups compared to that in both the CON and DKD groups (Fig. 1F). To detect ectopic lipid accumulations in the proximal tubules, Oil Red O staining and IHC of the lipid droplet biomarker ADRP was performed (*Heid et al., 1998*). The Oil Red O staining and ADRP, analysis revealed more lipid droplet in the DKD+L,

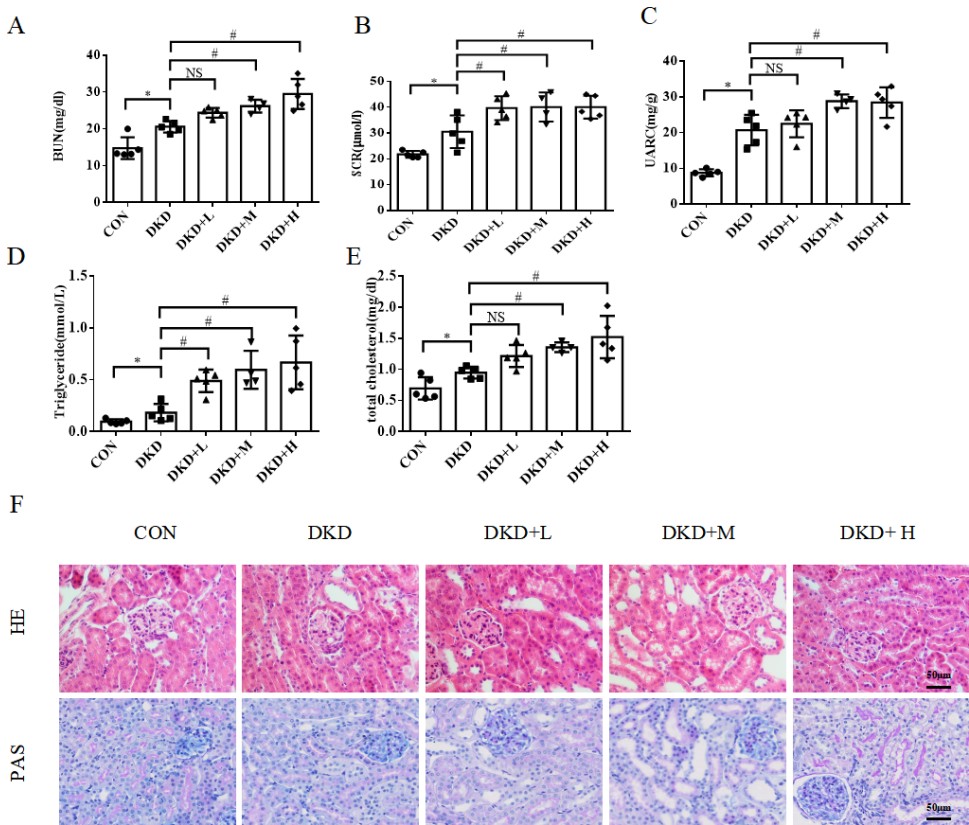

**Figure 1** **The effect of PM2.5 exposure on kidney function, serum lipid and kidney damage of mice.**
(A) Blood urea nitrogen, (B) serum creatinine, (C) urine albumin to creatinine ratio, (D) serum triglyceride, (E) serum total cholesterol in CON, DKD, DKD+L DKD+M or DKD+H groups. (F) The representative HE and PAS staining images in renal tissues from CON, DKD, DKD+L, DKD+M or DKD+H groups. Scale bars: 50 μm. Data are expressed as mean ± SD. * $p < 0.05$ *vs.* CON; # $p < 0.05$ *vs.* DKD.

DKD+M and DKD+H groups than in the DKD group (Fig. 2A). In DKD mice kidney, the level of SREBP-1 mRNA was increased compared to the CON group, while the level of ATGL mRNA was decreased compared to the CON group. The SREBP-1 mRNA level was higher in the DKD+M and DKD+H group compared to the DKD group, while the ATGL mRNA level was lower in the DKD+M, DKD+H group compared to the DKD group (Figs. 2G, 2H).

The expression of SREBP-1 was significantly higher in the DKD tubules. Notably, the extent of tubular SREBP-1 staining was significantly higher in the DKD+L, DKD+M and DKD+H mice than in the DKD mice and CON (Figs. 2A, 2B). Moreover, the expression of ATGL was decreased in the renal tubules of the DKD mice compared.to that in the CON mice. This alteration was even more marked in the DKD+L, DKD+M and DKD+H groups, which exhibited a significantly lower expression of ATGL expression than the DKD mice (Figs. 2A, 2C).
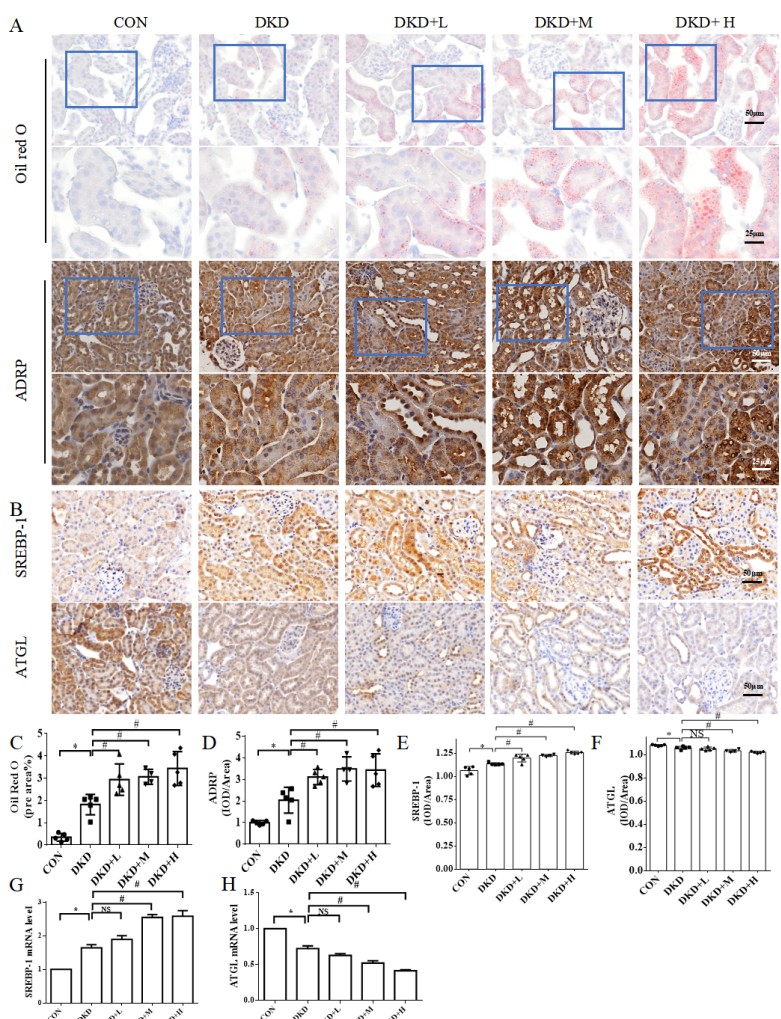

**Figure 2** **The effect of PM2.5 exposure on kidney ectopic lipid accumulation, lipid synthesis and lipi-dolysis of mice.** (A) The representative Oil Red O staining images and the representative immunohisto-chemical staining images of ADRP. High magnification of boxed areas is presented on the bottom. Scale bars: 50 μm (top), 25 μm (bottom). (B) The representative immunohistochemical staining images of SREBP-1 and ATGL in renal tissues from CON, DKD, DKD+L, DKD+M or DKD+H groups. Scale bars: 50 μm. (C) Quantification of Oil Red O staining. (D) Quantification of ADRP staining. (E) Quantification of SREBP-1 staining. (F) Quantification of ATGL staining. (G,H) qPCR analysis of SREBP-1 and ATGL mRNA levels in kidney tissues. Data are expressed as mean ± SD. * $p < 0.05$ *vs.* CON; # $p < 0.05$ *vs.* DKD.

## PM$_{2.5}$ exposure aggravates disordered lipid metabolism by upregulating the expression of HMGN1/KIM-1 in diabetic mice

The next analysis investigated whether PM$_{2.5}$ exposure causes kidney damage via HMGN1-KIM-1. HMGN1 and KIM-1 were detected by IHC. The levels of HMGN1 and KIM-1 in the tubules increased in the DKD mice compared with those in the CON. Moreover the extent of tubular HMGN1 and KIM-1 staining were significantly higher in DKD+L, DKD+M and DKD+H mice than in DKD (Figs. 3A–3C). There was significant correlation

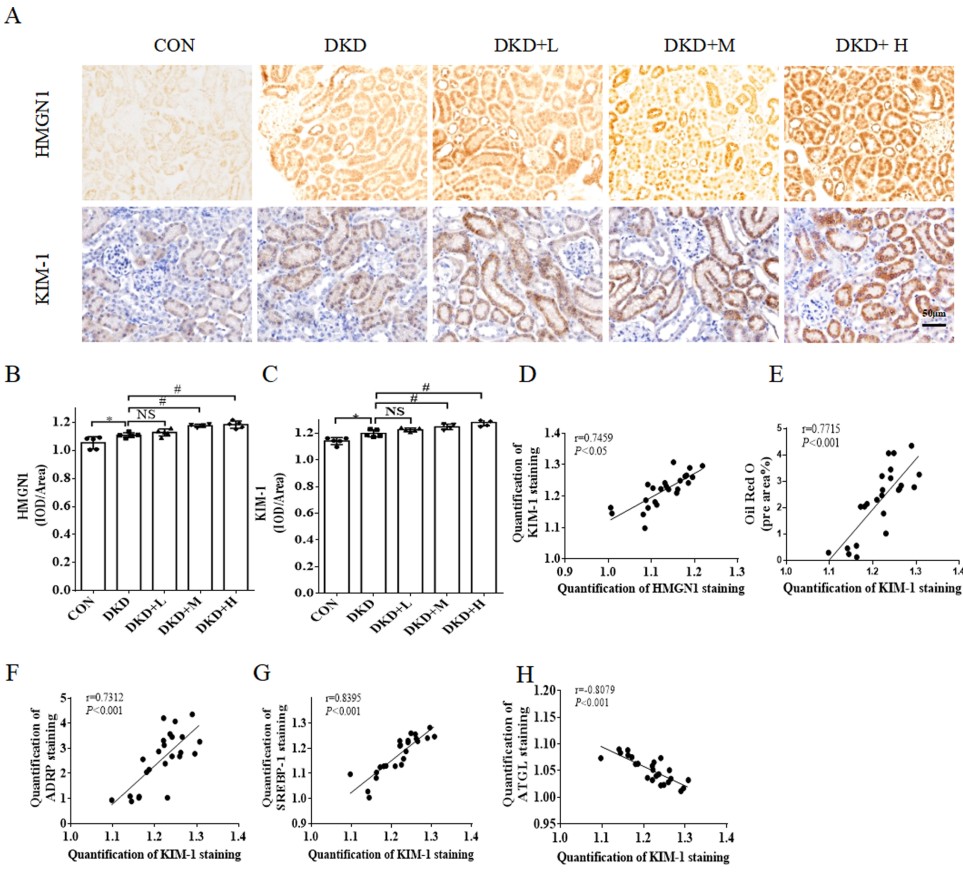

**Figure 3 The effect of PM2.5 exposure on kidney increases the expressiong of HMGN1 and KIM-1.** (A) Representative immunohistochemical staining images of HMGN1 and KIM-1 in renal tissues from CON, DKD, DKD+L, DKD+M or DKD+H groups. Scale bars: 50 μm. (B) Quantification of SREBP-1 staining. (C) Quantification of ATGL staining. Data is expressed as mean ± SD. (D) Correlation analysis between HMGN1and KIM-1. (E) Correlation analysis between KIM-1 and Oil Red O staining area%. (F) Correlation analysis between KIM-1 and ADRP. (G) Correlation analysis between KIM-1 and SREBP-1. (H) Correlation analysis between KIM-1 and ATGL. Data are expressed as mean ± SD. * $p < 0.05$ *vs.* CON; # $p < 0.05$ *vs.* DKD.

between the protein expression in the kidney KIM-1in the different groups and HMGN1. Positive correlations were detected between HMGN-1 and KIM-1 ($r = 0.7459$, $P < 0.001$) (Fig. 3D). The correlations between different groups Oil Red O staining area% and protein expression of ADRP, SREBP-1, ATGL were found to associate significantly with HMGN1. Specifically, positive correlations were detected between KIM-1 and Oil Red O staining area% ($r = 0.7830$, $P < 0.001$), ADRP ($r = 0.7312$, $P < 0.001$) and SREBP-1 ($r = 0.8395$, $P < 0.001$) (Figs. 3E–3G). Specifically, negative correlation was detected between KIM-1 and ATGL ($r = -0.8079$, $P < 0.001$).

## PM$_{2.5}$ exposure alters abundant metabolites profiles in the kidneys of diabetic mice

To systematically characterize the metabolic profiles in DKD and the potential metabolic alterations following exposure to PM$_{2.5}$ exposure, we performed a non-targeted metabolomic analysis of the s kidney tissue from the DKD and DKD+P groups. In total we identified 474 metabolites from the kidney samples (Table S1).

We constructed an OPLS-DA models for each sample. The models revealed the clustering and separation of the DKD and DKD+P groups (Figs. 4A, 4B; R2Y and Q2 for positive ion mode:0.994, 0.775; R2Y and Q2 for negative ion mode:0.995, 0.819). As demonstrated by the volcano plot, we further identified 50 significantly differentialy expressed metabolites (DEMs) between the DKD and DKD+P groups, based on the screening criteria of VIP > 1 and $P < 0.05$. Further analysis revealed a significant increase and decrease in the relative levels of 18 and 33 differential metabolites, respectively, after PM$_{2.5}$ exposure (Fig. 4C).

We submitted 50 DEMs to Emotionalist 5.0 for a KEGG pathway enrichment analysis. The DEMs from the kidney were significantly enriched in two pathways involving two types of pyrimidine metabolism and starch and sucrose metabolism ($P < 0.05$ and impact > 0.1; Fig. 4D). There were four DEMs in pyrimidine metabolism and uridine can cause massive ectopic lipid deposition increased in DKD+P (*Urasaki, Pizzorno & Le, 2016*). Dihydrouracil can improve lipid was downregulated in DKD+P (*Sullivan et al., 2022*). The expression of metabolites that promote lipid metabolism disorders increased: corticosterone and cortisone after PM$_{2.5}$ exposure (*Yang & Yu, 2021*; *Bai et al., 2022*), whereas the expression of metabolites that improve lipid metabolism: pantothenic acid, gentisic acid, dihydrouracil, pyridoxine and 3-Hydroxyanthranilate was downregulated in the kidney after PM$_{2.5}$ exposure (*Sullivan et al., 2022*; *Kang et al., 2021*; *Shibata et al., 2013*; *Danielyan & Simonyan, 2017*; *Berg et al., 2020*). These results are consistent with our biochemical findings and pathological staining detection of PM$_{2.5}$ aggravated kidney damage and kidney lipid metabolism disorder. In addition, several metabolites can be linked to oxidative stress and inflammation: pyridoxine, N-Acetylmannosamine, N-Acetylserotonin, and 3-Hydroxymethylglutaric acid (*Danielyan & Simonyan, 2017*; *Gao et al., 2022*; *Liu et al., 2021*; *Delgado et al., 2019*).

## DISCUSSION

In this study, a DKD+PM$_{2.5}$ exposure mice model was used to evaluate the effect of PM$_{2.5}$ on kidneys. We found that PM$_{2.5}$ aggravated kidney damage and abnormal lipid metabolism, which were correlated with increased expression of HMGN1 and KIM-1 expression. In addition, the LC-MS/MS analysis found an altered metabolic profile after PM$_{2.5}$ exposure in the kidney. These findings were consistent with previous research that PM$_{2.5}$ exposure increases metabolic disorders and kidney diseases.

We found that 8 weeks of PM$_{2.5}$ exposure resulted in metabolic syndrome, which is manifested by dyslipidemia. Meanwhile. Our analysis of PM$_{2.5}$-induced lipid metabolism indicators in mouse kidney tissue sections further confirmed the increased ectopic accumulation of lipids and renal damage in the kidneys of PM$_{2.5}$-exposed DKD mice.

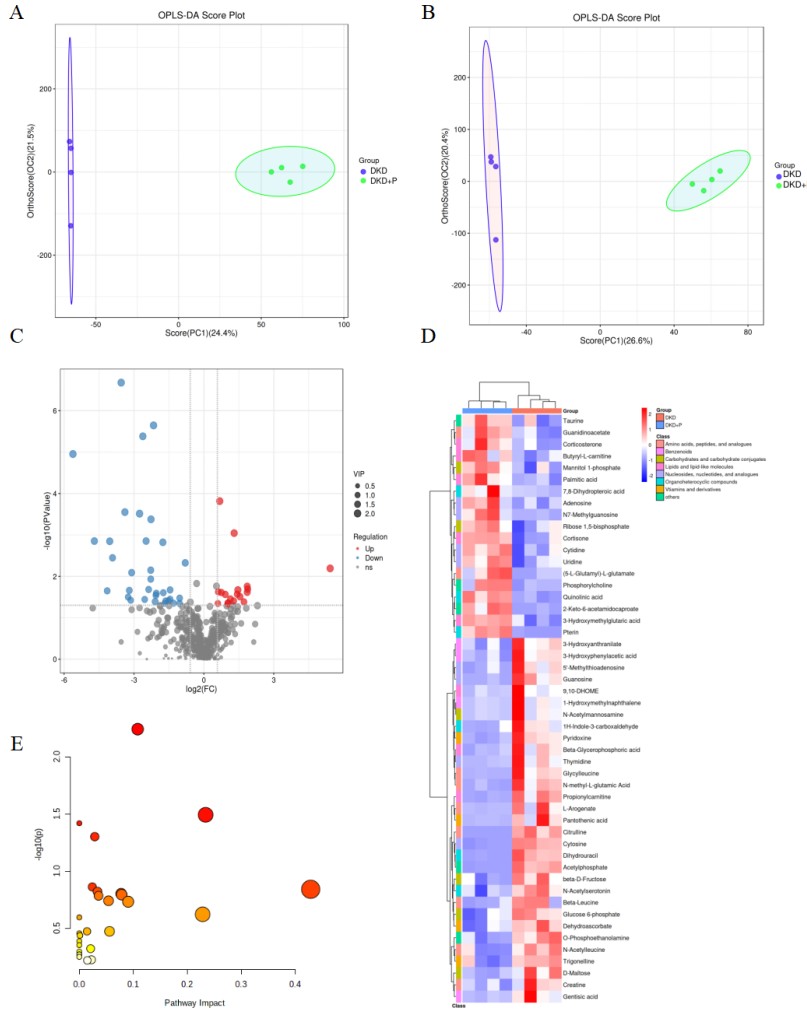

**Figure 4** **Effect of PM2.5 exposure on metabolites of kidney in mice.** (A) OPLS-DA score map for positive ion mode data. (B) OPLS-DA score map for negative ion mode data. (C) Volcano plot of the differential metabolites between the DKD and DKD+P group. (D) Heatmap depicting the expression levels of the 52 metabolites altered by PM2.5. Those 52 metabolites were classified for 8 categories according to the HMDB database. (E) Bubble diagram of enriched pathways of DKD and DKD+P mice.

$PM_{2.5}$ was further found to promote kidney injury and systemic metabolic disorder in a dose-dependent manner in the mice. Meanwhile, this study has shown that the histopathologic changes in the kidney and abnormal lipid metabolism became more common and severe with increasing dose of $PM_{2.5}$, consistent with earlier studies (*Xu et al., 2019*; *Zhang et al., 2017*). However, there was no statistical-significant difference among some of the indicators in different dose groups, the 8 weeks of $PM_{2.5}$ exposure may have been insufficient to reflect the effects of different $PM_{2.5}$ doses. This indicates that the results of this study should be reassessed over a longer duration.

Our previous study found that HMGN1 expression was elevated in DKD and kidney damage (*Yu et al., 2018*) and can cause increased KIM-1 expression (J Yu, J Da, F Yu, J

Yuan, Y Zha, 2022, unpublished data). The increase in HMNG1 expression after PM$_{2.5}$ exposure and the positive correlation between HMGN1 and KIM-1 expression were consistent with previously-reported findings. We also found that KIM-1 was also found to be positively correlated with lipid drops biomarker and lipid synthesis molecules, and negatively correlated with lipid decomposed molecules. It indicated that HMGN1-KIM-1 may aggravate the lipid metabolism disorder in the kidney. The causal relationship between them needs to be further investigated.

This study reports a significant increase in the expression of HGMN1, a well-known mediator in the pathogenesis of chronic low-grade inflammatory processes, in mice exposed to PM$_{2.5}$. Furthermore, alterations in metabolites, including N-Acetylmannosamine, Gentisic Acid, Citrulline, and 5′-Methylthioadenosine, which are all related to inflammation and oxidative stress. which have been associated with inflammation and oxidative stress, were detected. These findings align with prior research demonstrating that PM2.5 exacerbates oxidative stress and inflammation (*Diabetes & Air Pollution Collaborators, 2022*; *Wang et al., 2022*). These findings suggest that inflammation and oxidative stress may play a role in PM$_{2.5}$-induced kidney damage. Further research on this topic is important.

Fically, while the findings suggest a significant association between PM$_{2.5}$ exposure and alterations in lipid metabolism in the kidneys, further research is needed to fully elucidate the underlying mechanisms and potential confounding factors. Additionally, longer-time studies are required to better understand the long-term effects of these findings. Therefore, future research should aim to address these limitations and provide additional insights into the complex relationship between air pollution and kidney health.

## CONCLUSIONS

Our study provides the first experimental evidences that PM$_{2.5}$ caused abnormal lipid metabolism, kidney damage and changed metabolites (sugars, amino acids, etc.) in kidneys of DKD mice, the mechanism may be related to the increased expression of HMGN1-KIM-1. These results may provide new insight into our knowledge of the potential molecular mechanism by which PM$_{2.5}$ exposure increases the risk of kidney damage associated with metabolic disorder.

## ADDITIONAL INFORMATION AND DELCARATIONS

### Funding

This work was supported by grants from The Special Fund for Basic Scientific Research Operating of Central Public Welfare Research Institutes, the Chinese Academy of Medical Sciences (2019PT320003), the Guizhou high-level innovative talents program [QKHPTRC(2018)5636-2], the Guizhou Clinical Research Center for Kidney Disease (QKHPTRC[2020]2201), the Science and Technology Fund project of Guizhou Provincial Health Commission in 2020 (gzwkj2020-084), the Science and Technology Fund project of Guizhou Provincial Health Commission in 2021 (gzwkj2021-136), the Youth Fund of Guizhou Provincial People's Hospital in 2021 (GZSYQN[2021]12), and the Basic Research

Plan of Guizhou Province in 2022 (Natural Science Project) (QKH-ZK[2022] General 265). The funders had no role in study design, data collection and analysis, decision to publish, or preparation of the manuscript.

## Grant Disclosures

The following grant information was disclosed by the authors:

The Special Fund for Basic Scientific Research Operating of Central Public Welfare Research Institutes.

The Chinese Academy of Medical Sciences: 2019PT320003.

The Guizhou high-level innovative talents program: QKHPTRC(2018)5636-2.

The Guizhou Clinical Research Center for Kidney Disease: QKHPTRC[2020]2201.

The Science and Technology Fund project of Guizhou Provincial Health Commission in 2020: gzwkj2020-084.

The Science and Technology Fund project of Guizhou Provincial Health Commission in 2021: gzwkj2021-136.

The Youth Fund of Guizhou Provincial People's Hospital in 2021: GZSYQN[2021]12.

The Basic Research Plan of Guizhou Province in 2022 (Natural Science Project): QKH-ZK[2022] General 265.

## Competing Interests

The authors declare that there are no competing interests.

## Author Contributions

- Yuecheng Jiang conceived and designed the experiments, performed the experiments, prepared figures and/or tables, and approved the final draft.
- Yanzhe Peng performed the experiments, prepared figures and/or tables, and approved the final draft.
- Xia Yang conceived and designed the experiments, prepared figures and/or tables, and approved the final draft.
- Jiali Yu conceived and designed the experiments, authored or reviewed drafts of the article, and approved the final draft.
- Fuxun Yu analyzed the data, authored or reviewed drafts of the article, and approved the final draft.
- Jing Yuan analyzed the data, authored or reviewed drafts of the article, and approved the final draft.
- Yan Zha conceived and designed the experiments, authored or reviewed drafts of the article, and approved the final draft.

## Animal Ethics

The following information was supplied relating to ethical approvals (i.e., approving body and any reference numbers):

The animal protection and utilization committee of the People's Hospital of Guizhou Province approved the study.

## Data Deposition

Raw data are available Supplemental Files.

## Supplemental Information

Supplemental information for this article can be found online at http://dx.doi.org/10.7717/peerj.15856#supplemental-information.

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
