# Peer review of "PM2.5 exposure aggravates kidney damage by facilitating the lipid metabolism disorder in diabetic mice"

_PeerJ, doi:10.7717/peerj.15856_

## Round 0.1 · original submission · Major Revisions

Dear Jiang,

Thank you for submitting your manuscript to PeerJ. The referees have reviewed your manuscript carefully and recommended some modifications on the manuscript before further processing. Hence, the decision “Major revision” was taken for your submitted manuscript.

The referee would like to see easily the modifications made to your manuscript in the revised version. Therefore, I invite you to respond to the referee(s)' comments and revise your manuscript carefully.

Do not forget to highlight ALL the changes you make, using track changes.

Please provide also an answer/report to the referee(s)’ comments, which summarizes the changes you have made IN the manuscript itself. The answer/report to the referee(s) may also include any other response that you want the editor and the reviewer(s) to note. You should submit the answer/report to the referee(s)’ comments as a separate document.

Thank you for submitting your manuscript to PeerJ and giving us the opportunity to consider your work.

We look forward to receiving your revision.

Sincerely,

Reviewer 1 ·

Basic reporting

no comment

Experimental design

no comment

Validity of the findings

no comment

Annotated reviews are not available for download in order to protect the identity of reviewers who chose to remain anonymous.

Reviewer 2 ·

Basic reporting

Title: PM2.5 exposure aggravates kidney damage by facilitating the lipid metabolism disorder in diabetic mice

1. Introduction part is very limited and it does not provide enough information to understand further manuscript. Authors need to explain background, previous published literature and study rationale clearly. Please re-write this part.
2. No spaces between words. EX: Synthesisandlipolysis
3. Please add all abbreviations used in this study.
4. “Disorders of lipid metabolism are widely regarded as key factors in the occurrence and develoPMent of diabetic kidney disease “. This sentence must be revised.
5. Line 61-62, “increased synthesis gene expression” please revise it. Please go through all over the manuscript and revise necessary.
6. English writing must be improvised.
7. I encourage authors to study long PM2.5 exposures.
8. There is no strong mechanism to explain this manuscript other than some phenotype caused by PM2.5 exposure. Please discuss more about this part.
9. Please mention limitations of this study and possible explanation from author’s point of view.

Experimental design

I encourage authors to study long PM2.5 exposures also.

Validity of the findings

No comment

---

## Round 0.2 · accepted · Accept

Dear Dr. Jiang,

We are happy to inform you that your manuscript, which is entitled "PM2.5 exposure aggravates kidney damage by facilitating the lipid metabolism disorder in diabetic mice" has been Accepted for publication in PeerJ.

Thank you for your submission to PeerJ.

kind regards,

Reviewer 1 ·

Basic reporting

I am happy with the revisions made to the manuscript and the answers suffice my questions with this study

Experimental design

no comment

Validity of the findings

no comment

Reviewer 2 ·

Basic reporting

None

Experimental design

None

Validity of the findings

None

Additional comments

Authors answered as per reviewers comments one by one, that is appreciable. I would recommend this manuscript for publication. Before it goes for publication authors must correct few minor things Ex: Grammatical errors, font sizes of letters. This is important to understand the manuscript clearer.